# Phylogenetic and Structural Insights into Melatonin Receptors in Plants: Case Study in *Capsicum chinense* Jacq

**DOI:** 10.3390/plants14131952

**Published:** 2025-06-26

**Authors:** Adrian Toledo-Castiñeira, Mario E. Valdés-Tresanco, Georgina Estrada-Tapia, Miriam Monforte-González, Manuel Martínez-Estévez, Ileana Echevarría-Machado

**Affiliations:** 1Unidad de Biología Integrativa, Centro de Investigación Científica de Yucatán (CICY), Calle 43 No. 130, Mérida CP 97205, Mexico; atole291@gmail.com (A.T.-C.); georgina.estrada@cicy.mx (G.E.-T.); mmgcicy@gmail.com (M.M.-G.); luismanh@cicy.mx (M.M.-E.); 2Centre for Molecular Simulations, University of Calgary, Calgary, AB T2N 1N4, Canada; marioe911116@gmail.com

**Keywords:** melatonin, phytomelatonin receptor PMTR, G protein-coupled receptor, phylogeny, HMM, molecular docking, *Capsicum*, *Arabidopsis*

## Abstract

Recently, it has been proposed that plant melatonin receptors belong to the superfamily of G protein-coupled receptors (GPCRs). However, a detailed description of the phylogeny, protein structure, and binding properties of melatonin, which is still lacking, can help determine the signaling and function of this compound. Melatonin receptor homologs (PMTRs) were identified in 90 Viridiplantae sensu lato proteomes using profile Hidden Markov Models (HMM), which yielded 174 receptors across 87 species. Phylogenetic analysis revealed an expansion of PMTR sequences in angiosperms, which were grouped into three clades. Docking studies uncovered a conserved internal melatonin-binding site in PMTRs, which was analogous to the site in human MT1 receptors. Binding affinity simulations indicated this internal site exhibits stronger melatonin binding compared to a previously reported superficial pocket. Ligand–receptor interaction analysis and alanine scanning highlighted a major role of hydrophobic interactions, with hydrogen bonds contributing predominantly at the internal site, while non-interacting charged residues stabilize the binding pocket. Tunnel and ligand transport simulations suggested melatonin moves favorably through the internal cavity to access the binding site. Also, we presented for the first time details of these pockets in a non-model species, *Capsicum chinense*. Taken together, the structural analyses presented here illustrate opportunities and theoretical evidence for performing structure–function studies via mutations in specific residues within the proposed new melatonin-binding site in PMTRs, shedding light on their role in plant melatonin signaling.

## 1. Introduction

Melatonin (N-acetyl-5-methoxytryptamine) is an ancient molecule concomitant with life, with an old function as a major protectant against oxidative stress, which later extended to signaling in multicellular organisms [1]. Phytomelatonin, melatonin of plant origin, plays a role in maintaining circadian rhythms within plants [2] and is involved in various plant growth processes, such as lateral root formation, seed germination promotion, and ripening and senescence processes (see review [3]). Also, melatonin induces tolerance to stresses such as drought, salinity, and cold, among others [4]. Little evidence has been provided to explain the mechanism by which melatonin acts in plants [5]. However, melatonin may perform several of these functions via a receptor-dependent manner, as has been proposed in a recent and controversial field of study [6,7,8,9].

Melatonin receptors (MTRs) have been known in animals for at least 30 years [10]. These proteins are a small group of receptors that belong to the G protein-coupled receptor (GPCR) superfamily, which are receptors that have seven transmembrane domains (7TMRs). The two known melatonin receptors in humans, MT1 and MT2, are located in either the plasma membrane or mitochondria and are involved in melatonin-mediated synchronization of circadian rhythms and related physiological functions via GPCR signaling pathway [11,12]. The MT1 and MT2 receptor crystal structures in a complex with different agonists have already been determined, which allows us to understand the structural characteristics of the ligand pockets and establish the molecular bases of ligand recognition by melatonin receptor [13,14,15]. Knowledge of these structures led to the discovery of new chemotypes with new functions, which offered the opportunity to modulate the biology of these receptors and thus addressed problems of sleep disorders and depression in animals [16].

The identification of GPCR homologs in plants had initially been challenging due mainly to sequence divergence from those in animals [17]. However, profile Hidden Markov Models (HMM) have been successfully used to detect homologs in plants [18]. HMM are robust when the sequence divergence is not extreme, and there are sufficient positive controls. The appropriateness of its use is further supported by the increasing experimental evidence of plant melatonin receptors (PMTRs) in monocots and eudicots, which allows for more accurate predictions.

PMTRs were first reported by Wei et al. [6] in *Arabidopsis thaliana* (AtPMTR1), who proposed a module capable of inducing stomatal closure through the promotion of reactive oxygen species (ROS). Later, the role of PMTRs was further expanded to include responses against biotic [19] and abiotic [9] stress in *Nicotiana benthamiana* and *Medicago sativa*, respectively. Recently, a role played by these receptors, in which they mediate root regeneration and interaction with plant hormones, was reported [8]. To date, PMTR homologs to AtPMTR1 have been characterized in *Nicotiana benthamiana* [19], *Medicago sativa* [9], *Manihot esculenta* [20], *Zea mays* [21], *Oryza sativa* [22], *Gossypium hirsutum* [23], and *Solanum lycopersicum* [8].

Multiple sequences of PMTRs were identified within eudicot species, which differed at least in ligand binding capabilities [8,19,20]; however, only those closest in sequence similarity to Arabidopsis have been characterized [23]. So far, homologs have also been found in monocot representatives such as corn [21] and rice [22]. Although the melatonin binding capabilities of the receptors have been repeatedly demonstrated experimentally, little attention has been paid to where this interaction occurs and which structural determinants are involved. Recently, Barman et al. [22] suggested that PMTRs might bind melatonin at an external pocket, which is in marked contrast to known MTRs in animals. In addition, phosphorylation could be one of the post-translational modifications (PTMs) that tune the functional state of the receptor [20]. Increasing evidence supports PMTRs, located on plasma-membrane, being coupled to G-protein signaling in plants; however, data refuting this as a bona fide receptor exist [7].

In this study, we establish a stringent pipeline to retrieve putative PMTRs (pPMTRs) in representative species across Viridiplantae. Using these identified sequences, we investigate, with in silico tools, evolutionary features that set plant PMTRs apart from their animal counterparts, providing insights into their structural and, ultimately, functional divergence in plants. We also selected PMTR proteins from the habanero pepper (*Capsicum chinense* Jacq.) to expand our knowledge of their structural characteristics. This Solanaceae is one of the most economically and culturally relevant species grown in southeastern Mexico. This pepper is not only a highly valued condiment, but its metabolites have a wide range of industrial and pharmaceutical applications [24]. Although the national cultivation area of habanero peppers has increased to meet demand, problems such as soil erosion, low nutrient availability, salinity, and pests, among other factors, have led to reduced yields [24,25].

## 2. Results

### 2.1. Identification of Melatonin Receptor Candidates

Our homology search, using AtPMTR1 as a query in the HMMER server, yielded 833 entries. Following initial detection, 239 sequences met selection cut-off criteria based on *E*-value and protein length (Figure 1). Then, to ensure taxonomic diversity, sequences were grouped by species, and only the sequence with the highest similarity within a representative set of species was retained. Later, seven proteins previously reported and experimentally tested were added to a total of 15 sequences from the previous step. Finally, 16 sequences belonging to asterids (4), basal angiosperms (1), basal eudicots (2), fabids (3), malvids (2), vitales (1), and monocots (3) were used to build the HMM (Figure 1; Appendix A). In this dataset, protein length varied from 227 to 309 residues. Sequence alignment revealed a C-terminal domain highly conserved across taxa. Conversely, the amino-terminal region was markedly variable and was trimmed from the alignment to build the HMM. Forty-three percent of the chosen sequences to build the HMM matched with previously reported and experimentally tested PMTRs. This suggests that our selection strategy was adequate in identifying relevant homologs.

A total of 90 plant proteomes representing major phylogenetic groups were screened using previously built HMM to identify PMTR homologs. Across the proteomes, 253 candidate proteins were initially identified (Appendix A). After applying criteria *E-value* and sequence-length thresholds, 47 sequences were excluded, resulting in 206 candidates. A partial sequence from *Gossypium hirsutum* (NCBI: XP_040951491.1) that lacked a starting methionine was further removed. Additionally, 13 sequences with fewer than seven predicted transmembrane domains were discarded, along with 18 redundant entries. Following these filtering steps, 174 PMTR homologs were identified across 87 species, comprising ∼69% of the initial matches after applying inclusion criteria. In the microalgae *Chlamydomonas reinhardtii* and *Volvox carteri*, two and one sequences were detected, respectively, though none passed the filters established. Notably, no hits could be found in the magnoliid *Cinnamomum micranthum* for PMTR homologs.

Ancestral lineages like bryophytes and lycophytes both contained two sequences while marchantiales, ferns, cycads, and basal angiosperm *Amborella trichopoda* presented only one (Figure 2). *Glycine max* and *Musa acuminata* showed the highest number of sequences with six each. Most species included in this study contained two sequences (47.1%), while 34.5% of the species had only one sequence and 18.4% more than two sequences. Monocots generally presented one copy. In contrast, at least two copies were observed in most eudicots.

### 2.2. Phylogeny of Plant Melatonin Receptors

A maximum-likelihood tree was built from a multiple sequence alignment (MSA) of the 174 PMTR homologs found in this study (Figure 3). The branching structure agrees with the species tree previously obtained. In addition, the evolutionary tree evidences an early split between the distant marine green algae *Micromonas commoda* and the rest of Viridiplantae.

**Figure 1 plants-14-01952-f001:**
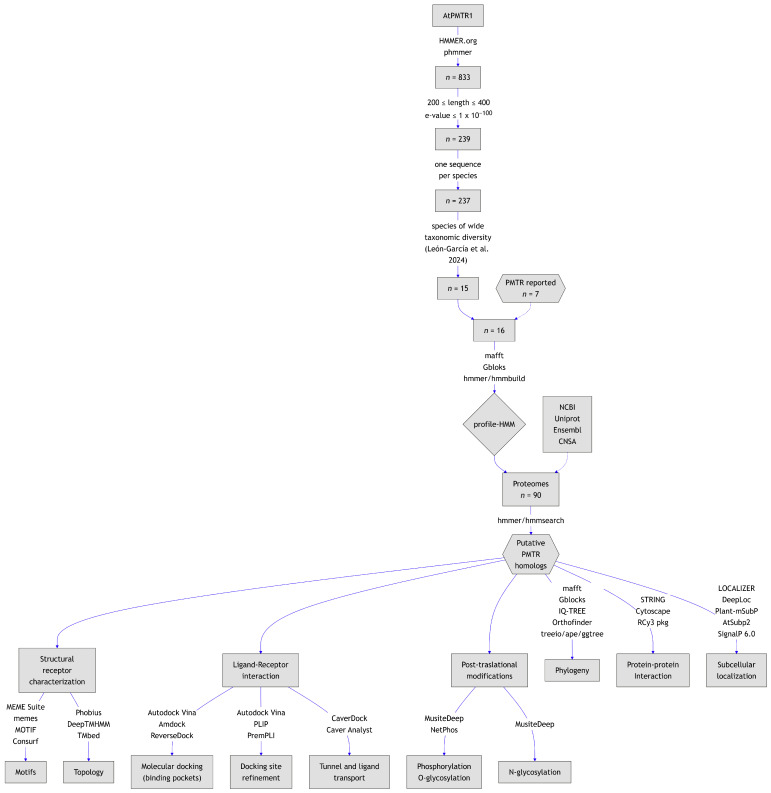
Graphical summary of the workflow followed in this study. Species spanning diverse taxonomic groups were selected based on León-García et al. [26] to build the HMM.

**Figure 2 plants-14-01952-f002:**
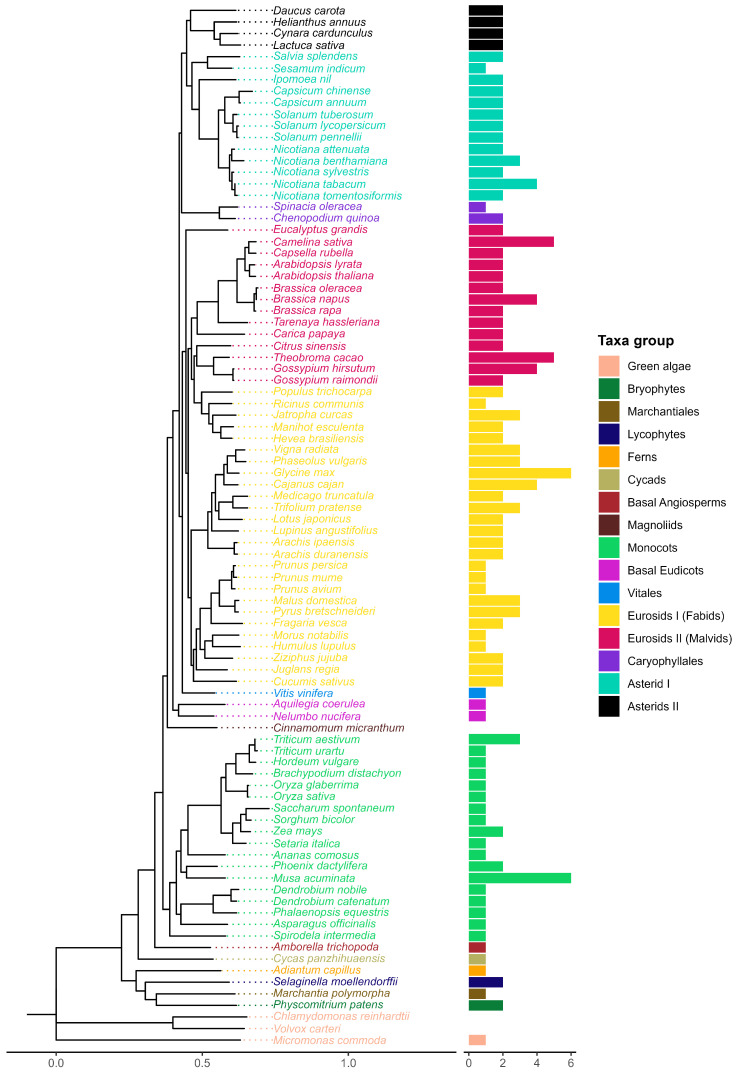
Evolutionary tree of the species included in this study (**left**). A consensus-rooted species tree was inferred by Orthofinder [27] from 90 plant proteomes. Horizontal bars (**right**) represent the number of sequences per species from a total of 174 sequences identified after the screening process.

**Figure 3 plants-14-01952-f003:**
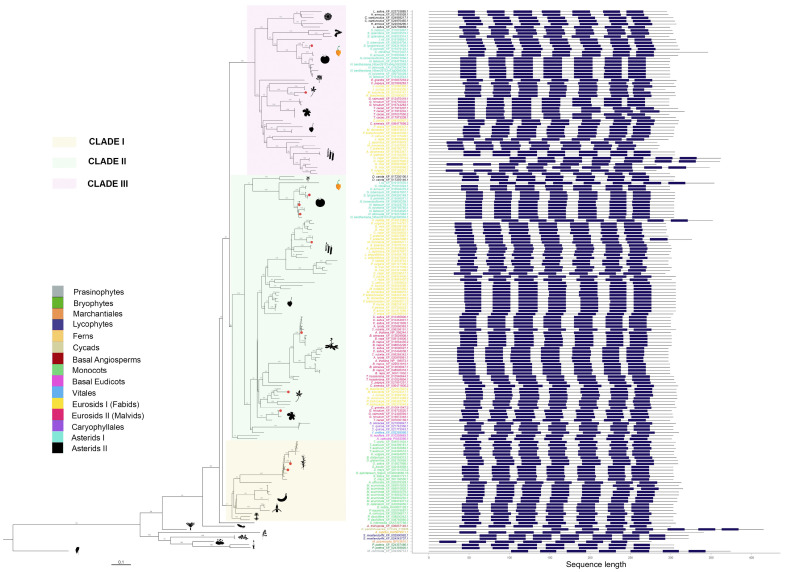
Phylogenetic tree of putative melatonin receptors for 87 species across 39 plant families. Numbers along branches represent support values obtained by 1000 bootstraps. Angiosperm representatives were grouped into three clades, which are highlighted. Taxa groups were mapped into colors. Images representing well-known taxa group representatives were included. Protein topologies obtained with Tmbed [28] are represented on the right side of the tree, with transmembrane regions represented as dark blue bars. Red dots in the branch tips represent those melatonin receptors previously reported.

pPMTR sequences from non-vascular plants were clustered together and separated from vascular plants. Angiosperms were grouped into three clades. Clade I comprises monocots clearly separated from eudicots with a total of 26 sequences distributed among 17 species. Eudicots are subdivided into two sister clades, Clade II, which groups a total of 57 species with 85 sequences, and Clade III, with 40 species and 53 sequences. Interestingly, all sequences from members of Vitales, Caryophyllales, and Basal Eudicots are only found in Clade II. While most eudicots containing more than one sequence were distributed between clades II and III, *Daucus carota* was the unique member of Asterids II with two sequences clustered in Clade II. In contrast, the sequences belonging to the rest of Asterids II, including *Lactuca sativa*, *Hemianthus annuus*, and *Cynara cardunculus*, were exclusive to Clade III. This is the first report describing pPMTRs in *C. chinense*, with two members: CcPMTR1 (PHU10024.1) found in Clade II while CcPMTR2 (PHU15407.1) grouped in Clade III (Figure 3).

All 174 pPMTR candidates were predicted to contain seven transmembrane domains flanked by N- and C-terminal domains. The N-terminal domain (NTD) of the analyzed sequences was, on average, ∼25% longer than the C-terminal domain (CTD). The NTD lengths ranged from 12 to 151 residues, while the CTD lengths varied from 26 to 68 residues. Among the sequences, *Marchantia polymorpha* (BFI23610.1) showed the shortest NTD, and *Cycas panzhihuaensis* (CYCAS_019808) had the longest. No significant differences in the lengths of CTDs or NTDs were observed across major taxonomic groups or clades. However, an unusually long NTD was noticed in *Cajanus cajan* (XP_020227919.2, XP_020227920.2) within Fabids, while CcPMTR2 exhibited an exceptionally long CTD, compared to CcPMTR1 and the rest of sequences from angiosperms included in the study (Figure 3).

Transmembrane domains were spaced by three extracellular (ECL) and three intracellular loops (ILC). ECL3 is the shortest in predicted length among these loops, varying from six to 12 residues with a median of eight residues. ECL2 was the longest loop across all the sequences analyzed with a median of 16 residues. Notably, two *Theobroma cacao* (XP_017973204.1, XP_017973207.1) sequences had unusually long ECL2 domains. Interestingly, ECL2 and ECL3 predicted lengths were consistently shorter in members of Clade III when compared to the other clades (*p* <0.05). Furthermore, ICL1 and ICL3 predicted lengths were longer in monocots (*p* <0.05) (Appendix A).

### 2.3. PMTR Protein Family: Motif Analysis

Plant PMTRs has been commonly related to canonical human MT1 and MT2 receptors but also to transmembrane protein adipocyte-associated 1 (TPRA1). PFAM and InterPro databases suggest this relationship as these proteins are members of the TPRA1/Protein CANDIDATE G-PROTEIN COUPLED RECEPTOR 2 (CAND2)/CAND8 family (InterPro: IPR018781 (https://www.ebi.ac.uk/interpro/entry/InterPro/IPR018781/, accessed on 10 July 2024), PFAM: PF10160 (https://www.ebi.ac.uk/interpro/entry/pfam/PF10160/, accessed on 10 July 2024)). The most conserved motif of pPMTRs was YYSEM[KR]D (Figure 4 and Figure 5). This motif, located at the distal end of CTD, was presented in 173 proteins except for the one from *Adiantus capillus* (KAI5079511.1). Similarly, the DFFxE[ED] motif located right after TM7 seems to be highly conserved. A CHG motif in the transition NTD-TM1 was mostly conserved in clades I and II (Figure 5 and Appendix A). The C35^1.30At^ within this motif in TM1 and C99^2.61At^ in TM2 form a disulfide bond which stabilizes the tridimensional structure. The second cysteine can be found in motif QxWEC (Figure 5). Also, an LEhS motif, where *h* stands for hydrophobic residue, is a greatly conserved sequence with key amino acids as E122^3.50At^ and S124^3.52At^ 100% conserved (Figure 4). Motif searches using tomtom against Eukaryotic Linear Motifs (ELM) (http://elm.eu.org, accessed on 1 September 2024) and PROSITE (https://prosite.expasy.org/, accessed on 1 September 2024) databases yielded a few statistically relevant hits (Appendix A). However, the annotated motif sequences showed substantial divergence from the queries used, and their associated biological functions were primarily restricted to animals, with rare occurrences in plants.

Among the transmembrane domains, TM3 was the most conserved in sequence, while TM6 showed the lowest, with no fully conserved residues across the sequences analyzed. Tryptophan residues in the extracellular half of TM2, TM3, and TM5 were 100% conserved in the studied sequences (Appendix A). In sequences from seed plants, ICL1 contained a conserved S71^1.66At^ and ICL3 was the largest intracellular loop, enriched in proline with a conserved region LPx[RK] (Appendix A). Interestingly, within ECL2, a highly variable region, only a F171^5.38^ was found to be conserved across all the protein sequences studied.

**Figure 4 plants-14-01952-f004:**
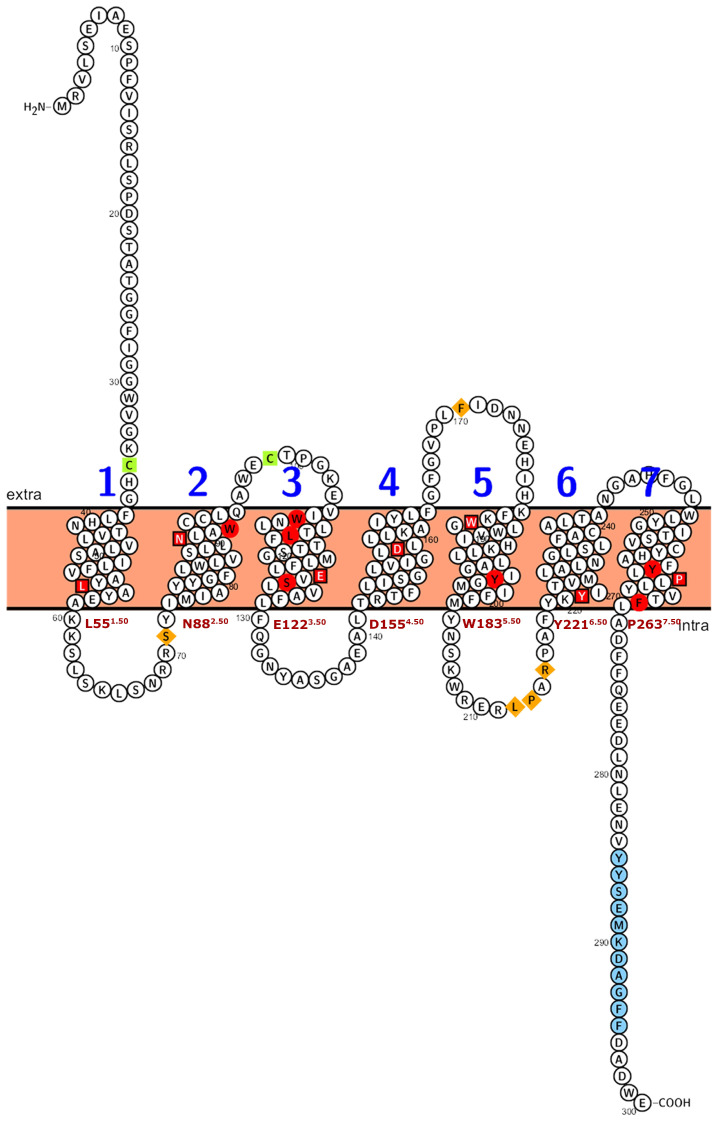
Numbering scheme for PMTRs. The most studied PMTR, AtPMTR1, was used as a reference to highlight topology and key conserved residues across 174 pPMTR sequences. Fully conserved residues are shown in red, mostly conserved in ECL and ICL domains in spermatophytes in orange, and cysteines involved in disulfide bridges in green. The most conserved residue for each TM domain, depicted as a squared box (light font color), was determined by consensus between our results and ConSurf [29]. These residues serve as the reference for the proposed PMTR numbering system based on Ballesteros and Weinstein [30]. Briefly, the first letter denotes the amino acid followed by its position in the protein, and then, in superscripts, the identifier starts with the TM number and ends with its position relative to the reference residue in that TM. That reference residue is arbitrarily assigned the number 50 [30]. The proposed reference residues are shown in red letters below each TM domain. YYSEM[KR]DAGFF is a motif completely conserved in seed plants (light blue). Domain predictions were derived from Tmbed [28] and visualized using PROTTER v1.0 [31].

**Figure 5 plants-14-01952-f005:**
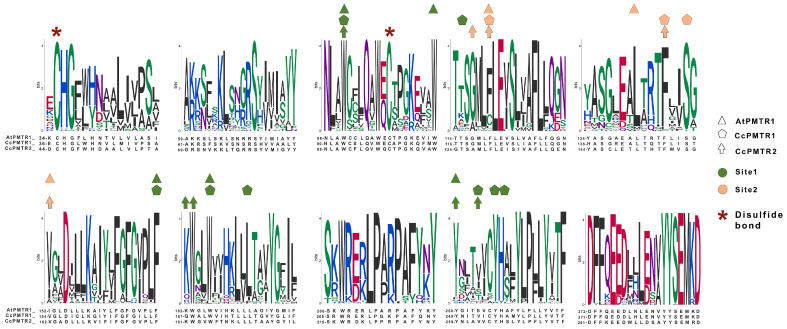
Motifs discovered by MEME from the alignment of the 174 PMTR homolog sequences. A; Residues that interact with melatonin in site 1 and site 2 in AtPMTR1, CcPMTR1, and CcPMTR2 are represented with colors and shapes. Cysteine residues involved in disulfide bonds are highlighted. Residue coloring follows universalmotif package [32] defaults.

### 2.4. Subcellular Localization

Signal peptides were only detected for *M. commoda* (XP_002499712.1, Pr: 0.99) and two Poaceae representatives, *Setaria italica* (XP_004981721.1, Pr: 0.69) and *Brachypodium distachyon* (XP_003559312.1, Pr: 0.55). Localization to subcellular compartments, as indicated by the tools used in this study, is summarized in Appendix A. DeepLoc consistently predicted all proteins to be localized at the plasma membrane. However, predictions from other tools varied across clades. LOCALIZER indicated that 19.2% of proteins in Clade I were targeted to mitochondria, whereas Clades II and III exhibited plastid localization in 2.4% and 5.7% of proteins, respectively. This trend was further supported by Plant-mSubp, which predicted plastid localization for 11.5% of Clade I proteins, in contrast to Clades II and III, where 60% and 70% of proteins were assigned to the same location. Meanwhile, AtSubP2 suggested that both *A. thaliana* proteins were likely localized to the plasma membrane.

We selected the most known PMTR, AtPMTR1, and two newly found pPMTRs, CcPMTR1 and CcPMTR2, in order to characterize the in-depth structural and functional features of these plant receptors.

### 2.5. Post-Translational Modifications

AtPMTR1, CcPMTR1, and CcPMTR2 exhibited a similar distribution of predicted phosphorylation sites, with a higher frequency of serine phosphorylation sites near the NTD and intracellular loop 1 (ICL1) domains (Appendix A). In contrast, phosphorylated tyrosines were identified within a conserved motif in the C-terminal region of all three receptors. Notably, N-glycosylation sites were exclusively predicted for CcPMTR1. NetPhos identified 18, 21, and 26 phosphorylated sites for AtPMTR1, CcPMTR1, and CcPMTR2, respectively. However, there was limited agreement between PTM prediction tools for AtPMTR1 and CcPMTR2 (Appendix A).

### 2.6. Molecular Docking Predictions for Melatonin Binding to PMTRs

Melatonin has been proposed to interact with AtPMTR1 within a superficial pocket delimited by TM2, TM3, and TM4 [22]. To investigate this interaction in receptors of *A. thaliana* and *C. chinense,* we performed exhaustive blind docking simulations using Autodock Vina. PMTRs from both species apparently share two distinctive binding pockets (Appendix A). However, these sites differed by the frequency in which they occurred in the simulations. When visually inspected, the more frequent pocket appeared to be located in the previously reported region (site 2), while the other seemed to be located internally, embedded within the TM domains (site 1). Although site 1 was less frequent, the binding affinities observed in this first approach suggested a better binding for this site. Afterward, we used the ReverseDock server to try to corroborate these findings. Surprisingly, this program failed to detect the novel site, site 1, in these three receptors. Based on the previous findings, we speculated that the blind docking parameters used might not have been the most appropriate. Therefore, we adopted the AMDock tool workflow, which enabled us to systematically explore PMTRs previously reported as well as homologs across the taxa groups included in our study (Figure 6). The analysis revealed that both sites were consistently identified in most proteins across Viridiplantae (Figure 6). *M. polymorpha* pPMTR1 (MpPMTR1) bound to melatonin at ECLs showed low binding poses restricted to tridimensional regions with low quality structure predictions. Receptors from ancestral bryophytes and lycophytes, along with the more recent basal angiosperms, also presented site 1 as a probable interaction pocket for melatonin. In contrast, cycads and monocots lacked a clearly defined pocket, and their binding scores were generally lower compared to those of eudicots. Both *A. thaliana* PMTRs exhibited a similar distribution of binding pockets and melatonin affinities. Site 1 was broadly conserved among clades II and III members, with binding scores suggesting a higher affinity for melatonin in clade III. Notably, proteins from Asterids II exhibited a strong interaction with melatonin restricted to site 1 with minimal evidence of additional binding sites (Figure 6).

We used the inactive conformation of hMT1 crystal structure (PDB: 6me2) (https://www.rcsb.org/structure/6me2, accessed on 10 October 2024) as an internal control to test for the docking simulation accuracy. Initially, hMT1 (Uniprot: P48039) (https://www.uniprot.org/uniprotkb/P48039/entry, accessed on 10 October 2024) modeled with ColabFold [33] and the crystal were structurally aligned with a resulting root mean square deviation (RMSD) of 0.48. Then, Amdock [34] was able to identify the internal pocket using the predicted structure as input. Additionally, the molecular docking with melatonin in the orthosteric binding site produced poses similar to that of the crystal (Figure 6). This suggests that the docking method was able to make accurate predictions for pockets and poses.

To better estimate binding scores, we proceeded with a probabilistic approach through 1000 simulations at the previously identified pocket coordinates for sites 1 and 2 (Figure 7A). In general, site 1 exhibited lower binding scores compared to site 2 for both *A. thaliana* and *C. chinense* receptors indicating, a higher affinity for melatonin in this site. Binding energy analysis revealed comparable affinities across PMTR homologs: AtPMTR1 (−7.32 kcal·mol^−1^ [95% CI: −7.36, −7.27] at site 1; −6.20 kcal·mol^−1^ [−6.26, −6.15] at site 2), CcPMTR1 (−7.26 [−7.30, −7.21]; −6.22 [−6.26, −6.19]), and CcPMTR2 (−7.45 [−7.50, −7.40]; −6.55 [−6.67, −6.44]). Ligand efficiencies obtained for melatonin in these receptors were approximately 0.43 and 0.37 kcal·mol^−1^ for sites 1 and 2, respectively. Interestingly, the distribution of poses found after the simulations suggests that site 1, the buried pocket, is more restrictive with narrower bell-shaped distributions in the receptors evaluated. In contrast, distributions coming from site 2, the superficial pocket, were skewed toward higher energy conformations (Figure 7A).

In general, hydrophobic interactions were the most frequent interaction type in both sites, accounting for ∼60% of all the interactions detected. In site 1 higher binding values could be associated to a higher contribution from hydrogen bonds. However, with no directly interacting residues, the presence, in site 1, of the AtPMTR1 of the H36^1.31^ located in the NTD, Q95^2.57^ and E98^2.60^ in ECL2, and H258^7.45^ in TM7 could be especially important and possibly stabilizes the 3D structure (Figure 7E). In particular, computationally calculated E98^2.60^ mutation by alanine was extremely unfavorable to ligand–receptor complex stability. Additionally, F171^5.38^ in ECL2 and Y250^7.37^ in TM7 form hydrogen bonds to the indol ring and amine group in the melatonin tail (Figure 7B1). F80^2.42^ and L146^4.41^ were the most influential residues interacting through hydrophobic interactions in site 2 of AtPMTR1 (Figure 7B2). Furthermore, F113^3.41^, S116^3.44^, G149^4.44^, and L156^4.51^ were relevant but not due to a direct interaction with ligand.

**Figure 6 plants-14-01952-f006:**
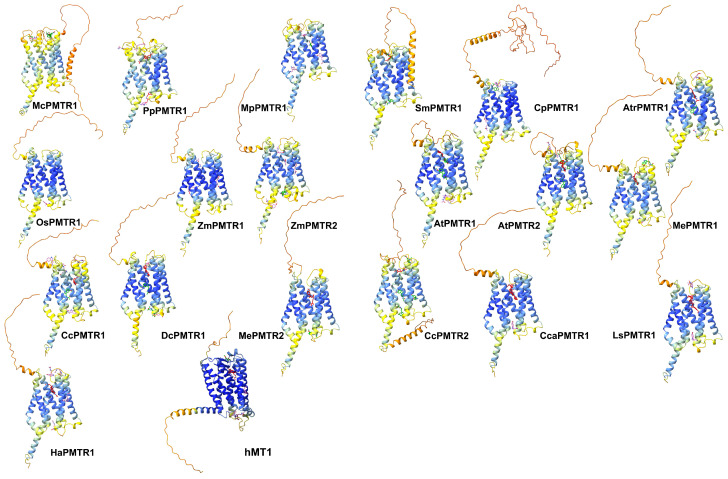
Putative docking sites for melatonin in PMTRs across Viridiplantae. Ligand poses obtained by AMDock [34] are presented for: *M. commoda* (McPMTR1), *P. patens* (PpPMTR1), *M. polymorfa* (MpPMTR1), *S. moellendorffii* (SmPMTR1), *C. panzhihuaensis* (CpPMTR1), *A. thaliana* (AtPMTR1, AtPMTR2), *O. sativa* (OsPMTR1), *Z. mays* (ZmPMTR1, ZmPMTR2), *A. trichopoda* (AtrPMTR1), *M. esculenta* (MePMTR1, MePMTR2), *C. chinense* (CcPMTR1, CcPMTR1), *Cynara cardunculus* (CcaPMTR1) *D. carota* (DcPMTR1), *L. sativa* (LsPMTR1), *H. annus* (HaPMTR1), and hMT1. Receptors are presented as cartoons where colors represent pLDDT scores for the predicted structure (dark blue: very high, blue: high, yellow: low, orange: very low). Ligand poses were classified based on binding affinity and colored accordingly as red (∆G≤−7 kcal·mol^−1^), green (−7<∆G≤−6 kcal·mol^−1^), and pink (∆G>−6 kcal·mol^−1^).

Our simulations showed that CcPMTR1 binds melatonin in site 1 in a deeper position compared to AtPMTR1 (Figure 7C1). In CcPMTR1, site 1 exhibited a pronounced sensitivity to E124^3.50^ missense mutation located on ECL2, despite this residue not directly interacting with melatonin (Figure 7E). Moreover, Y256^7.44^ in CcPMTR1 forms π−π stacks with the indol ring of melatonin. Alanine substitution at this position also had a substantial impact on interaction. In contrast, in site 2, most interacting residues contributed mainly through hydrophobic interactions with only minor destabilization of the ligand–receptor complex during the mutational scan (Figure 7C2).

**Figure 7 plants-14-01952-f007:**
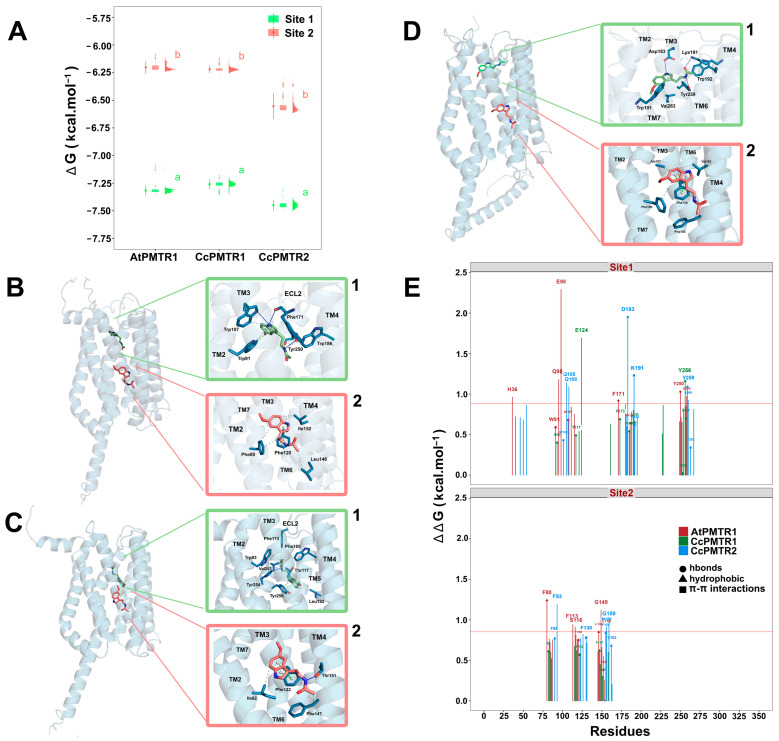
Binding pockets and ligand–receptor interactions for PMTRs of *A. thaliana* and *C. chinense*. (**A**), binding affinity distributions obtained from one-thousand Autodock Vina docking simulations for MT and receptors: AtPMTR1, CcPMTR1, and CcPMTR2. Site 1 (green) and site 2 (pink) for each receptor are shown, as well as the mean with its 95% confidence interval, boxplot, and histogram. A sample (n = 5) from the simulations was used to compare mean binding affinities between sites for each species independently. Distinct letters denote significant differences identified at a 95% confidence level using a Student’s *t* test. Receptor cartoon representations bound to melatonin in site 1 (green MT) and site 2 (pink MT) with ligand–receptor interactions for site 1 (superior green box, 1) and site 2 (inferior pink box, 2) for AtPMTR1 (**B**), CcPMTR1 (**C**), and CcPMTR2 (**D**). Hydrogen bonds (dark blue solid lines), π-π stacks (dashed green lines) and hydrophobic interactions (dashed grey lines) are shown. TM1-7, transmembrane domains; ECL2, extracellular loop 2. (**E**), residue relevance in the ligand–receptor interaction obtained through alanine-scanning and PLIP interaction profiler for AtPMTR1 (red), CcPMTR1 (green) and CcPMTR2 (blue). The horizontal red line is drawn at the 90th percentile.

In CcPMTR2, the polar residues Q105^2.57^ and Q108^2.60^ and N260^7.38^ were influential in site 1 but not due to direct interaction with the ligand (Figure 7E). Instead, the charged residues D183^5.41^ and K191^5.49^ participated in hydrogen bond formation with the ligand (Figure 7D1). Not surprisingly, these residues contributed markedly to melatonin positioning within the embedded pocket. Residues Y250^7.37^ in AtPMTR1 and Y259^7.37^ in CcPMTR2 were positionally homologous within TM7 and also contributed with hydrogen bonds to stabilize the melatonin aliphatic tail. F93^2.45^ in CcPMTR2 was the corresponding residue to F80^2.42^ in AtPMTR1, both located in site 2, but it was not directly involved in ligand binding (Figure 7D2).

### 2.7. Tunnel and Ligand Transport Analysis

Since site 1 is an internal pocket in the protein, it is important to know whether it presents a tunnel through which melatonin can access the site. To explore this possibility, tunnel and cavity analyses were performed. AtPMTR1 contained an internal cavity (site 1) of 1023.0 Å^3^ that can be accessed through a tunnel with dimensions of 8.2 Å in length and 1.5 Å in bottleneck radius and throughput 0.79 located among TM1, TM2, and TM7. However, this tunnel apparently showed an entrance partially embedded in the lipid bilayer, which is probably exposed to the extracellular space (Figure 8A). Interacting residues along the ligand trajectory through the tunnel are shown in Figure 8A, where D173^5.40^, E98^2.60^, and H36^1.31^ residues are located at the tunnel entrance (Appendix A). Transport analysis suggested melatonin could move through the tunnel favorably with negative binding affinities (Figure 8D). Additionally, the estimated energy at the binding site was somehow lower than what was previously predicted by simulations.

**Figure 8 plants-14-01952-f008:**
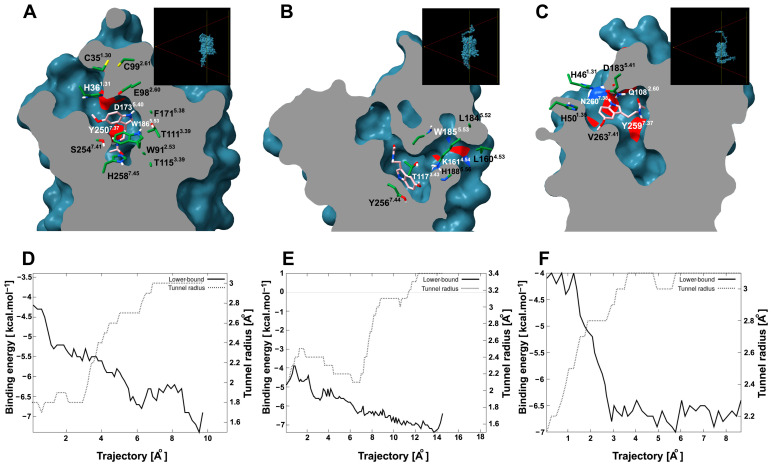
Tunnel and ligand transport analysis for internal melatonin binding site 1 in *A. thaliana* and *C. chinense*. Receptors bound to melatonin were sliced longitudinally at the ligand plane, which exposes the internal cavities and tunnels: (**A**), AtPMTR1; (**B**), CcPMTR1; (**C**), CcPMTR2. Black boxes in the top-right corner display a lateral view of the cutting plane. Receptor residues (green) interacting with melatonin (pink) during its transit are represented as sticks. Tunnel radius and binding energy during melatonin trajectory through the tunnel for AtPMTR1, (**D**); CcPMTR1, (**E**); CcPMTR2, (**F**).

CcPMTR1 presented the largest cavity found with 1654.4 Å^3^ in volume. A tunnel 13.7 Å long and 1.8 Å bottleneck radius was found in this receptor with a lateral entrance located between TM4 and TM5 within the lipid bilayer (Figure 8B). On the other hand, CcPMTR2 cavity was 1141.1 Å^3^ in volume. TM2, ECL2, and TM7 surround the channel opening, which faces the extracellular space (Figure 8C). The most biologically relevant tunnel was shorter in length (7.2 Å) but with a wider bottleneck radius (2.0 Å) than previous homologs. Relevant residues neighboring the tunnel access were H188^5.56^ in CcPMTR1 and D183^5.41^ and N260^7.38^ in CcPMTR2 (Figure 8B,C).

### 2.8. Protein–Protein Interaction

STRING networks integrate experimental evidence and text-mining data to predict interactomes and identify potential functional modules. AtPMTR1 was closely related to CAND3, CAND6, and CAND7—all CAND family members. It was also predicted to link to GCR2, a debated ABA receptor, through experimental validation, but probably to a Lanthionine synthetase C (LanC) family member instead. Hyp1, a Pathogenesis-Related 10 (PR-10) protein from *Hypericum perforatum* capable of weakly binding up to two melatonin molecules, was also identified. Not surprisingly, physical interaction with the $G_{\alpha }$ subunit and a functional association with acyl serotonin methyl transferase (ASMT), the enzyme that catalyzes the rate-limiting step in the melatonin biosynthesis pathway in plants, were predicted (Figure 9A).

**Figure 9 plants-14-01952-f009:**
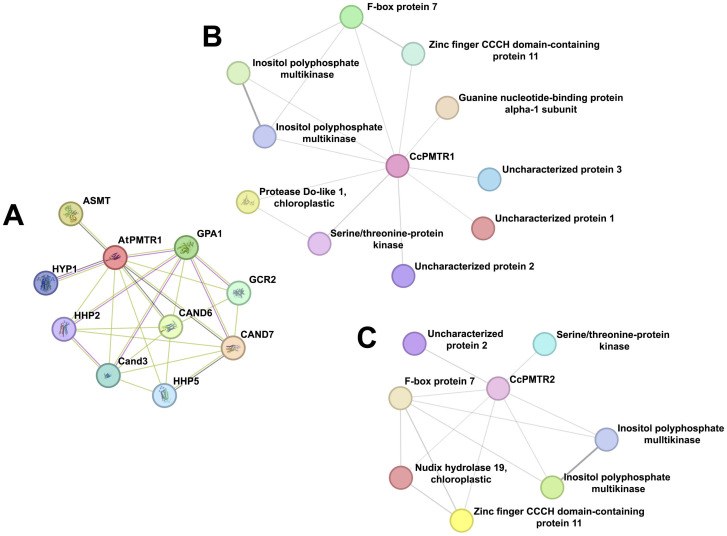
Protein–protein interaction networks for PMTRs in *A. thaliana* and *C. chinense*. (**A**), AtPMTR1; (**B**), CcPMTR1; (**C**), CcPMTR2.

The network analysis suggested that CcPMTR1 might interact with kinases, a chloroplastic protease, F-box protein, zinc finger CCCH-type domain protein, G-protein alpha 1 subunit (GPA1), and uncharacterized proteins 1–3 (Figure 9B, Appendix A). Also, interactions of CcPMTR1 with two types of kinases that could mediate the activation of independent signaling transduction pathways were predicted. Uncharacterized proteins 1, 2, and 3 appeared to be limited to the *Capsicum* genus as evidenced using a blastp search. CcPMTR2 shared most of the interaction partners predicted for CcPMTR1. Notably, chloroplastic Nudix hydrolase 19 was a unique partner for this protein compared to its paralog. The Nudix superfamily (IPR000086) comprises pyrophosphohydrolases that act upon substrates of general structure NUcleoside DIphosphate linked to another moiety (Figure 9C, Appendix A). Remarkably, no G-protein subunit was predicted to interact with this protein.

## 3. Discussion

### 3.1. Origin of PMTRs

Melatonin is an ancient molecule distributed specially among primitive bacteria [35]. During the transition to an oxygen-rich atmosphere, melatonin may have pivoted from a primary scavenger molecule into a signaling molecule mediating ROS homeostasis [2]. The origin of PMTRs is probably situated at some point ranging from 7TMRs started to diverge in plants and metazoans (∼1.6 BYA) [17] and the red and green algae separation (∼1.2 BYA) [2].

Probably, as a consequence of this divergence, several initial studies struggled to identify plant GPCR homologs. Early success in the identification of candidates from this family resulted from the development of stringent bioinformatic pipelines that considered the weaknesses and strengths of state-of-the-art tools for protein classification [17,18]. To date PMTRs have been identified and characterized in monocots [21,22] and eudicots [6,8,9,19,20,23]. In monocots, only one sequence per species has been reported, while most eudicots contained two, which is consistent with our results. Furthermore, the distinctive separation between clade II and clade III could be associated, but not exclusively, to differences in binding to melatonin. This separation was achieved thanks to we included in this study ancestral species within each taxon, which were not considered in previous studies [22]. All these sequences have been retrieved using canonical blastp searches; however, the lower sensitivity [36] compared to HMM-based methods may contribute to missing potential relevant candidates. That is the case in AtPMTR2 and ZmPMTR2, which were reported in this study but not detected previously.

A remarkable finding was that the expansion of pPMTR sequences in eudicots may have consequences beyond receptor affinity. Recent crown members from Asterids II seem to have transitioned from the most ancestral balance in sequences in clades II and III toward expanding clade III sequences at the expense of clade II. Unexpectedly, *D. carota* seems to have a contrasting trend retaining only and expanding sequences belonging to clade II, which we could not yet explain. In later studies, a major number of Asterids II members should be included to corroborate this preference.

Diversification and expansion in PMTRs could also be a way to respond to different signals and stress. In the solanaceae *N. benthamiana*, homologous to AtPMTR1 mediated salicylic acid accumulation in leaves, which contributed to systemic resistance against *Phytophthora nicotianae* infection [19]. Another solanaceae, tomato, contains a member on both clades II and III. Apparently, SlPMTR2 expression seems to be targeted to roots in contrast to its homolog from clade II, which shows a more widespread distribution [8]. This suggests that the more recent clade III member may have evolved to play tissue specific functions during evolution. Another consistency found in tomato and cassava was that it appears that receptors from clade III seem to bind melatonin less strongly than clade II proteins [8,20]. In cotton, Zhang et al. [23] detected about eight putative members from the CAND2 family. These members showed marked differences in their expression levels in response to salt, drought stress, and heat, and were organ specific. Additionally, only one sequence, GhCAND2-D5 (D11G2677.1), was upregulated by both salt and exogenous melatonin application [23], which corresponded to NCBI: XP_016753520.1 (https://www.ncbi.nlm.nih.gov/protein/XP_016753520.1, accessed on 27 February 2025) in our study. In corn, ZmPMTR1 expression levels depended on stimuli such as hydrogen peroxide and hormones, salinity, and extreme temperatures, and this response was tissue dependent [21]. In addition, some effects mediated by PMTRs, even those belonging to the same clade, show functional differences between species. In alfalfa, MsPMTR1 fully recovered seed germination under salt stress consistently across several varieties, and this receptor was able to respond to serotonin, besides melatonin [9]. In contrast, AtPMTR1 seems to mediate seed germination inhibition in normal conditions in a dose-dependent manner after melatonine treatment [37,38,39]. In summary, PMTR expansion in angiosperms might suggest diversification in ligand range and binding affinity, response to internal or external stimuli, and tissue or organ specific responses.

Freshwater microalgae *C. reinhardtii* respond to melatonin application by inducing anti-oxidant enzymes under nitrogen-starvation, reducing stress and promoting lipid-accumulation [40,41]. Microalgae are able to synthetize melatonin, particularly in response to stress, where it seems to act primarily as an antioxidant [42]. However, melatonin receptor homologs have not been reported in green algae so far. Interestingly, in our study, only the marine microalgae *M. commoda* had a putative homolog that, even after further examination, could not be detected in other freshwater and marine microalgae species. Additionally, some plant species like *Cinnamomum micranthum* may have lost this receptor. Exogenous melatonin application in another member of this genera, *C. camphora* seems to increase heavy metal accumulation through a phloem-mediated transport for soil decontamination [43]. pPMTRs under the criteria defined in this study are not present in this genus. If melatonin acts as the only antioxidant molecule or signal via another protein in this species, this could be elucidated in future research.

Using a distance-based method, Wei et al. [6] were able to demonstrate divergence from animal MTRs and a split between three monocots (only from Poales) and 11 eudicots (only from Rosids and Fabids). Bai et al. [20] showed that plant sequences clustered as monocots and eudicots also using a non-evolutionary model. Additionally, they reported that MePMTR2 localized externally from a clade grouping others receptors, including MePMTR1 [20]. Li et al. [2] reported an unrooted distance-based tree including a total of 14 representative species spanning major plant taxa from green algae to angiosperms. These previous results, though less broad in species diversity, are in agreement with our findings separating clades II and III. Our study provides a comprehensive and balanced representation of taxa across Viridiplantae, which allowed us to infer an evolutionary rooted tree consistent with current taxonomy.

Initial and subsequent findings support that melatonin can induce stomatal closure in multiple species [2,6,9,21] ameliorating drought and salt stresses. Intriguingly, a basal species as *M. polymorpha* have a pPMTR sequence, but lacking stomata is an interesting model to explore ancestral roles of PMTRs.

### 3.2. Subcellular Localization

Through in silico analysis, we suggested for the first time that pPMTR members belonging to phylogenetically distinct clades might have potentially distinct subcellular localizations, in addition to the prediction of the localization in the plasma membrane for all of them. Some proteins from clade I, which grouped the monocots, were tagged to mitochondria, whereas those from clades II and III were predicted to localize to plastids. Our results also highlight that most of these proteins lack a signal peptide.

Despite the contrasting research of Lee and Back [7], which reported the possible cytoplasmic localization of AtPMTR1, there is compelling evidence that this protein is localized to the plasma membrane [6,8,44]. Yu et al. [9] reported plasma membrane localization of MsPMTR1 after agroinfiltrating *N. benthamiana* leaves with the GFP-tagged receptor. Following a similar approach, Wang et al. [21] used Arabidopsis seedlings expressing ZmPMTR1 fused to GFP, reaching similar conclusions. Bai et al. [20] also corroborated this finding and further demonstrated colocalization of MePMTR1 and MePP2C1, a phosphatase regulating receptor functionality, at the plasma membrane using a bifluorescence complementation assay in onion epidermal cells. Recently, the plasma membrane subcellular localization of SlPMTR1/2 proteins was reported by using transiently overexpressing GFP-fusion proteins in tomato protoplasts [8].

Melatonin concentration varies dramatically across species, but particularly among plant organs [45]. Under these circumstances, a correlation between PMTR affinity for ligand and melatonin content may be difficult to establish. A differential receptor targeting specific plastids may help to cope with environmental and physiological cues more specifically. Additionally, differences in receptor-ligand binding could further support this subcellular sorting, with low-affinity receptors targeted to where the accumulation of melatonin is higher, and vice versa.

Tan et al. [46] suggested mitochondria and chloroplasts were the original sites of melatonin synthesis in the early stage of endosymbiotic organisms, and their melatonin biosynthetic capacities have been preserved during evolution. However, the transition from the direct role of MT as antioxidant to a signal molecule associated to a receptor is not clear. Mitochondria-synthetized melatonin served as a signal to stimulate MT1 receptors located at the outer or inner membranes in animals, which was coined as automitocrine regulation [11]. These authors proved that the signal triggering the response was melatonin exclusively synthesized in the mitochondrial matrix acting through its organelle-localized receptor. It would be interesting to study whether this regulation also occurs in plants and whether the differential localization of these receptors between contrasting species, monocots and eudicots, leads to distinctive functions between them. Nowadays, it is well supported that GPCRs located on both mitochondrial inner and outer membranes modulate the physiological function of mitochondria in animals [12]. Additional findings support that GPCR signaling occurs in the secretory pathway as well [47]. However, these subcellular localization patterns remain little explored in plants.

Chloroplasts might be the main site where biosynthesis of melatonin occurs in plants [48]. However, at least three routes for melatonin biosynthesis exist in chloroplast and mitochondria that can be basally active or preferentially activated under stress [49]. Thus, plant plasticity is also increased with differential expression of enzymatic isoforms in response to internal or environmental cues [50].

### 3.3. Post-Translational Modifications

Exploring the possible post-translational modifications of PMTRs is relevant since it can regulate the biological function of the receptor, not only affecting its melatonin-binding properties [20], but also the subcellular localization of these proteins [51,52].

Both phosphorylation and glycosylation were predicted as probable post-translational modifications of AtPMTR1, CcPMTR1, and CcPMTR1 proteins in this study. Although experimental evidence of PTMs of PMTR is practically absent, Bai et al. [20] demonstrated that melatonin-delayed dark-induced leaf senescence is abolished when the phosphatase MePP2C1 dephosphorylates MePMTR1 at S11. These authors demonstrated that the phosphorylation in S11 residue of *M. esculenta* melatonin receptor is necessary for binding to a melatonin receptor. S11 was flanked by E and P (ESP), and this context was conserved for AtPMTR1, with S10 being the equivalent in AtPMTR1, which was predicted to be phosphorylated. However, the two pepper proteins did not have this conserved sequence, although several Ser near this position in the NTD were predicted to be phosphorylated. Our analysis suggests that probably an equivalent phosphorylation in NTD may also play a role in *A. thaliana* and *C. chinense*, highlighting that the poor conservation in this region could lead to differential regulations between species.

Also, some phosphorylations were predicted in CTD of three proteins. Barman et al. [22] through in silico analysis predicted residues to be phosphorylated in the CTD, ICL1, and ICL2 in both AtPMTR1 and OsPMTR1, though this is not yet experimentally demonstrated. Based on sequence alignment we found a S71^1.66At^ in the ICL1 that could be a potential phosphorylation target conserved across spermatophytes. Specific reversible phosphorylations in ICL domains have been shown to produce human receptor variants with distinct patterns of interaction [53]. Fully conserved consecutive tyrosines and a serine in the CTD of the PMTRs found in our study could be putative phosphorylation sites. Interestingly, Sethi et al. [54] showed that melatonin treatment-induced MT1 and MT2 internalization was probably mediated by a phosphorylation of Y295^7.64^ located in the transition to CTD.

A glycosylation site was predicted in NTD for OsPMTR1 [22], as reported in this study for only pepper proteins. N-linked glycosylations in TM, ECL, and NTD of tomato SlPMTR1/2, which is another solanaceae as pepper, were also recently predicted [8]. Although experimental evidence for this type of post-translational modification is lacking for plant receptors, N-linked glycosylations at the NTD have been associated to decreased expression [51] and differential subcellular trafficking [52] in animal GPCRs.

Most crystallographies available for GPCRs were obtained after modifications in the NTD and CTD with the removal of glycosylation and phosphorylation sites [15]. These changes may play a major role in the efforts to obtain the first PMTR crystals.

### 3.4. Structural Basis of Melatonin Binding and Receptor Activation

The binding pocket of human MTRs (hMTRs) are nearly identical but just a critical amino acid may determine ligand specificity and agonist/antagonist behavior toward the receptor [55]. In spite of the high divergence in sequence between hMTRs and PMTRs, the topology of the binding pocket is similar, allowing for melatonin accommodation in the internal cavity. To date, the molecular determinants of melatonin-PMTR interactions have just recently been investigated exclusively through in silico tools [8,19,22]. Kong et al. [19] reported AtPMTR1 Asn100 participates in ligand binding through hydrogen bonding interaction. Unfortunately, we observed that this residue is inconsistent with the canonical AtPMTR1 sequence.

Okamoto et al. [13] experimentally demonstrated a diminished MT1- inhibitory G subunit ($G_{i}$) signaling in the F179A and Q181E mutants. These residues located in ECL2 of MT1 and its homologs in MT2 are relevant in melatonin binding [55,56]. In plants, the only conserved residue in this loop is the homolog residue F171^5.38^, which is conserved in spermatophytes and suggests a functional and/or structural relevance. This was substantially corroborated by our in silico mutational scan, docking simulations, and evolutionary conservation analysis.

Differences in similarity among clades may reflect variation in the binding to melatonin. MePMTR1 (Manes.09G160300) from Clade II and MePMTR2 (Manes.14G085000) from Clade III showed similarities to AtPMTR1 in the amino acid sequences of 68% and 58%, respectively [20]. In this case, Bai et al. [20] demonstrated that in *M. esculenta* the melatonin binding ability of MePMTR1 is similar to AtPMTR1, but both are higher than that of MePMTR2. Similarly, Liu et al. [8] demonstrated through microscale thermophoresis that SlPMTR1 bound melatonin with higher affinity than SlPMTR2. These observations could not be corroborated by our docking simulations because Vina scoring cannot resolve differences below ∼1 kcal·mol^−1^ as estimated in this study from Clade II and III proteins (Figure 6). Since the prediction methods used in this study may have limitations and experiments to demonstrate ligand–receptor binding properties have been performed with heterologous proteins in non-physiological contexts, more data from in vivo studies and with a larger number of proteins may be necessary to clarify these discrepancies.

Additionally, Wang et al. [21] reported that ZmPMTR1 had strong binding ability with melatonin tested in vitro through microscale thermophoresis ($EC_{50}$ 47.8 ± 6.45 nM) but lower to that reported for AtPMTR1 ($K_{d}$ 0.73 ± 0.10 nM) by Wei et al. [6]. This agrees with our finding that eudicot PMTRs showed higher binding affinity scores compared to those of monocots, particularly Clade III members. Notably, ZmPMTR1 was able to rescue Arabidopsis mutants lacking the ortholog receptor AtPMTR1 even under stress conditions [21]. This indicates that despite differences in sequence and affinity between monocots and eudicots receptors, PMTR signaling is probably well conserved in plants.

Kong et al. [19] found that PMTR homologs in *N. benthamiana* with the lowest similarity to AtPMTR1 among the putative receptors identified in that study were actually the ones involved in stomatal closure. Unfortunately, sequences reported by the authors failed to match with annotated proteins present in neither *N. benthamiana* nor *A. thaliana* from our study, which makes it impossible to spot differences between clades. Additionally, reported data, including receptor length or residues involved in ligand–receptor interaction, were inconsistent with those from canonical AtPMTR1 reported by Wei et al. [6].

Wang et al. [21] using a blast search in corn could only detect by homology and subsequently study ZmPMTR1 but not its paralog ZmPTMR2 (Figure 3). While previous studies have employed similar procedures to identify pPMTRs, our results demonstrate that HMMs provide a higher resolution for detecting homologs even across divergent sequences. Paralogs are commonly associated with new functions, CAND8 (Uniprot: F4K2U8 (https://www.uniprot.org/uniprotkb/F4K2U8/entry, accessed on 27 February 2025), AtPMTR2) share an 83% identity with AtPMTR1 and might be a putative receptor that remains under-explored. In that sense, further investigation is required to experimentally establish functional differences among paralog PMTRs within the species.

Kong et al. [19] were the first to investigate a potential binding site for melatonin in PMTRs from *N. benthamiana*. However, although the binding pocket found appears to be at the receptor surface, they failed to report its specific location. More recently, Barman et al. [22] described a potential superficial binding site shared by both, *O. sativa* and *A. thaliana*, delimited by TM2,TM3, and TM4. To date there are no reports of superficial binding sites in this region for any ligand within the well-characterized Class A GPCR family in animals. Instead, all published evidence to date strongly supports an internal binding pocket within the TM domains in animal MTRs [13,16,56,57,58]. Most importantly, the validity of an external site in plants must be considered cautiously, given the limitations of blind docking simulations even after proper selection of relevant pockets.

Barman et al. [22] also reported Autodock Vina scores for melatonin interaction with both OsPMTR1 (AF-Q75J22) and AtPMTR1 in the external site (site 2, in this study). They found similar binding free energies were -5.9 and -6.1 kcal·mol^−1^ for rice and thale cress, respectively. These binding scores fall below the confidence interval computed in this study for Vina docking scores at the equivalent AtPMTR1 site. They reported that S116^3.44^ contributed with a hydrogen bond, and hydrophic interactions were the most relevant to stabilize receptor-ligand complex, which is partially consistent with our results.

We found melatonin binds the receptor in site 1 at different depths depending on the receptor (Figure 7). This may play a role in accommodating melatonin derivatives or even other signal molecules. This variability in the depth of ligand penetration into the TM bundle among different ligands but also across different receptors in the Class A GPCR family is well documented [53]. Further complexity to this interaction arises from an interesting phenomenon known as “biased ligands”, where ligands behave preferentially agonist toward a signaling pathway whilst antagonist to others [59].

The currently accepted activation mechanism for animal MTRs involves the following steps: when melatonin binds to the orthosteric site, key residues within conserved activation motifs—FIP, CWxP, NRY, YPYP, and NAxxP—undergo a rotational shift (Figure 10) [13,60]. This conformational change triggers an outward displacement of TM6, ultimately leading to the dissociation of the $G_{\alpha \beta \gamma }$ heterotrimer. Interestingly, the magnitude of this TM6 displacement varies among receptors and seems to be associated with the type of $G_{\alpha }$ subtypes interacting. Additionally, differences in the receptor cytoplasmic cavity are another putative determinant for the $G_{\alpha }$-coupling selectivity [13].

Our analysis failed to identify conserved FIP, CWxP, or YPYP activation motifs in PMTRs. However, we observed that the internal binding site in plants is enriched with fully conserved tryptophan residues (Figure 4) that may undergo structural rearrangement upon melatonin binding, potentially contributing to receptor activation. Variation across activating motifs within Class A GPCRs occur. For example, in hMTRs, the N124^3.49^R125^3.50^Y207^5.58^ motif deviates from the DRY motif, a consensus within the family. A similar triad in plants, E122^3.50At^S124^3.52At^Y197^5.64At^ where glutamate and tyrosine residues form a conserved hydrogen bond (∼1.6 Å) in distance) that could be destabilized after ligand binding may also participate in signaling. Y261^7.48At^xP263^7.50At^xxY266^7.53At^ motif which is the closest in sequence to the canonical NPxxY and Na^+^ binding site motif in animal GPCRs than to NAxxY in hMTRs. However, what stands out is that TM6 in PMTRs is the most variable domain in contrast to hMTRs and were the conserved CWxP domain stands. Thus, its role as the motive transmembrane domain mediating the GTP-exchange factor activity of the receptor in plants is difficult to establish.

Missing motifs that participate in animal GPCRs activation and the fact the $G_{\alpha }$ subunits also diverge functionally from those of animals obscure the mechanisms by which PMTRs might transduce signals. Bradford et al. [61] proposed that self-activating G protein probably represented the ancestral state, which later evolved toward a slow nucleotide exchange. This shift, accompanied by a co-evolution with 7TMs receptors, may have facilitated the radiating expansion of this signaling system in animals [61]. Interestingly, plant and animal $G_{\alpha }$ subunits diverge notably in the primary sequence, but the three-dimensional structure remains remarkably conserved [61]. Indeed, when structurally aligned AtGPA1 against human $G_{\alpha q}$ the RMSD is ∼5.9 Å. In contrast, AtPMTR1 and hMT1 RMSD are approximately three times higher. Such differences contribute to the ongoing debate surrounding G-protein–GPCR signaling mechanisms in plants [62,63].

### 3.5. Tunnel and Transport

All members of Class A GPCRs, also known as Rhodopsin-like receptors, including melatonin receptors, share a characteristic embedded site within the TM bundle. Nevertheless, the orthostheric binding site location at the extracellular half of the TM domains, the cavity topology, depth, and access to the binding sites varies markedly [64].

However, in the above reported site 1 in plants, TM2, TM3, TM4, and ECL2 are the most frequent interacting domains, with TM3 being additionally the most conserved in sequence. These results agree with previous studies, which refer to the pivotal importance of TM3 in GPCR structure, ligand binding, and receptor activation [53].

Melatonin is an amphipatic molecule that can diffuse the plasma membrane. Interestingly, animal melatonin receptors have a 7–11 Åburried entrance, measured from the outer border, into the PM [15,65]. Noteworthy, ECL2 blocks access to the binding site from the extracellular space in MT1 [15]. However, it seems to be common in class A GPCR structures that these extracellular domains may either obstruct the entrance to the binding site or allow a water-accesible entrance, acting as a “lid” [53,66]. For example, MT1 and MT2 crystal structures both show a lateral ligand entrance, but only MT2 reveals a narrow opening to the extracellular side [14,67]. Experimental evidence of a putative extracellular entry path was provided by Johansson et al. [14] using [^3^H]melatonin in radioligand binding assays. They tested several mutations: (1) to widen the ECL2 opening or (2) to obstruct the lateral entrance in MT1 and MT2, and demonstrated a drastic decrease in ligand residence time compared to the wild type, supporting the feasibility of an extracellular route in MT2. More recently, Bedini et al. [68], who used 2-iodomelatonin, a labeled melatonin agonist, and dynamic simulations, found that although an alternative unbinding route toward the extracellular region was possible for the MT2 receptor, the preferred path with a lower free-energy barrier was the lipophilic path. They demonstrated that mutation of Y294^7.39^, a residue obstructing the path, to alanine significantly increased the relevance of the former.

Based on that evidence, we suggest that something similar may happen in PMTRs and that the uniqueness of a single entrance to site 1 may be questionable. The existence of multiple entrances may provide an advantage to answer to signals: (1) from the apoplast mediating locally or long-distance cell-to-cell communication, (2) access through passive diffusion to cytoplasmic interactors, and (3) diffusion inside-out as a means of internal signaling or “autocrine”-like. Meanwhile, differences in the chemical nature of the ligand may lead to an alternative preferred route [53].

The binding pocket in MT1 is located within a cavity about 710 Å^3^ in volume [15]. Similar or higher volumes were estimated for cavities found in PMTRs for site 1 (Section 2.7). The canonical entrance to this cavity is a lateral tunnel. Stauch et al. [15] showed that A158^4.56^M mutation abolishes function in MT1 for all tested agonists because it blocks ligand entrance. Additionally, Y187/200^5.38^ (in MT1/MT2, respectively) may function as a “gatekeeper” of the lateral pore, adopting distinct conformational states that widen or narrow its aperture [65]. A similar site-directed mutagenesis approach targeting the most influential residues might help to unveil experimentally observed differences in PMTR tunnel entrance locations.

Our simulations are but an exploratory analysis to document the diversity in ligand–receptor interactions in PMTRs, which require a more in-depth study through molecular dynamic simulations and, ultimately, experimental evidence. Noteworthy, to date, no description of additional binding pockets for MT or other agonists in MT1 or MT2 has been reported. Which may probably indicate the low probability of the existence of alternative pockets. In our study, we failed to detect the entrance to the internal binding site in ZmPMTR1 and OsPMTR1, receptors that have been functionally tested. This may be a limitation of the predicted structures used for docking analysis. For example, it could be the case that the prediction corresponds to inactive states with a closed tunnel entrance.

### 3.6. Protein–Protein Interactions String Network

The search for regulatory networks in which PMTRs are involved is essential for predicting the diversity of biological functions in which these proteins participate. Receptors are known to participate in functional modules containing effector proteins such as kinases and scaffolding proteins [69,70]. These functional modules can regulate the activity of individual proteins and provide insight into their involvement in cell processes such as development and tolerance to environmental stress.

Experimental evidence has rigorously demonstrated PPI from signaling and regulation modules in PMTRs. Wei et al. [6] proposed a functional model for melatonin signaling through PMTRs, demonstrating co-localization of AtPMTR1 (AtCand2) with AtGPA1 and providing indirect evidence of their interaction via a yeast split-ubiquitin system, where growth in selective minimal media indicated physical association. This finding was further supported by Yang et al. [44] through co-immunoprecipitation analysis. Additionally, these authors reported melatonin-induced stomatal immunity in *A. thaliana* through AtPMTR1-mediated activation of MAPK kinases but AtGPA1-independent, supporting the existence of alternative signaling pathways. Furthermore, MePP2C1 phosphatase functions as a regulatory element through direct contact with MePMTR1; experimental evidence of the direct interaction was provided by Bai et al. [20] using a BiFC assay. Using a similar technique, Liu et al. [8] also confirmed the interaction between SlPMTR1/2 with SlGPA1.

CAND proteins and HHP2 were identified in the AtPMTR1 interaction network based on their computationally predicted and experimental supported associations with AtGPA1 [18]. All these proteins contain 7TM domains being, at least partially, related to GPCRs. Interestingly, GCR2 was probably misidentifed, and it is not structurally or functionally related to this family as multiple reports suggest [18,71]. Additionally, Hyp1 was probably identified due to its capability to weakly bind up to two melatonin molecules. However, term co-occurrence in cited literature extracted through text-mining suggests a functional link to ASMT.

In a quantitative trait loci (QTL) mapping study of tetraploid *Actinidia chinensis* var. *chinensis* for tolerance to *Pseudomonas syringae* pv. *actinidiae*, Tahir et al. [72] identified genes associated with plant immunity, including a homolog transmembrane protein like (PSS19622.1), which under the criteria defined in this study is a pPMTR, a Protease Do-like 1, chloroplastic, and Serine/threonine-protein kinase EDR1-like. Another predicted protein, Zinc finger CCCH domain-containing protein 11, contains a DFRP_C domain (https://www.ebi.ac.uk/interpro/entry/InterPro/IPR032378/, accessed on 20 November 2024) which modulates GTPase activity [73].

Other well-characterized partners in animal models might have a counterpart in plants that belong to the interactome of PMTRs. As previously stated, G-protein independent activation [74,75] proteins modulating its activity may play a major role in plants as a consequence. Regulator of G-protein signaling (RGS) acts by increasing GTPase activity of $G_{\alpha }$ proteins in plants [76,77]. Interestingly, pepper has an homologue to AtRGS1, a 7TM protein with RGS-box domain, CcRGS1 (NCBI: PHU03188.1 (https://www.ncbi.nlm.nih.gov/protein/PHU03188.1?report=genbank&log$=prottop&blast_rank=1&RID=Y5M6Z65W013, accessed on 20 November 2024)). Surprisingly enough, CcGPA1 lacks a canonical Rat sarcoma virus (RAS) domain which is associated to GTPase activity but contains an homologous conserved plant helical domain which confers to AtGPA1 its active basal state. Additionally, other PDZ domain-containing scaffolding proteins, distributed in bacteria and eukaryots [78], mediates G-coupling [79].

The existence of other signaling complexes should be considered, particularly if homologs of those proteins exist in plants, after copious experimental evidence from animals. For example, it is well supported that protein oligomerization plays a role in modulating MTRs activity [67]. MT1, MT2, and GPCR50 can form homo- and heteromers [59]. These complexes could bring positive or negative cooperativity through modulation of preferred ligand entrance to the binding site [80], changes in ligand binding affinity and selectivity [81], recruitment of internalization modules [82], and/or transactivation where one protomer binds the ligand, whereas the other interacts with the G protein [83,84].

Two results were interesting in relation to the interaction of CcPMTR2 with other proteins, which did not coincide with those obtained for CcPMTR1: the interaction with Nudix hydrolase 19 and the non-interaction with the G-protein subunit. It has been recently suggested that Nudix hydrolases participate in inorganic phosphate and iron homeostasis and in energy balance [85]. These enzymes have been linked to increased fungal pathogenesis by impairing inorganic phosphate sensing in plants, diverting attention from host [86]. This result, together with the probable existence of a G protein-independent receptor signaling pathway, may suggest functional divergences between both habanero pepper proteins, which were also grouped into different evolutionary clades in this study.

Knowledge generated in model species cannot always be applied to agricultural crops such as habanero pepper. For example, the response of the primary root of Arabidopsis to the exogenous presence of nitrogen sources such as nitrate and amino acids (glutamate) could not be replicated for habanero pepper [87,88,89]. When characterizing a high-affinity potassium transporter isolated from the root of habanero pepper, it was found to be insensitive to sodium, differing from its closest ortholog in *Capsicum annuum* [90], demonstrating that the results cannot be generalized even within the same genus. The study of the family of high-affinity nitrate transporters [91,92] and glutamate receptors [26] in habanero peppers highlights the unique characteristics of this species in responding to the specific environmental conditions in which it grows.

Although the habanero pepper has been a study model for various Mexican researchers since the 1990s [93], the background reinforces the importance of generating more basic knowledge about this species. The results of these studies can be used to design strategies that contribute to increasing the yields and quality of habanero peppers grown under salinity, nutrient deprivation, pathogen attack, and other stress conditions. These receptors are proteins that can intervene in the signaling pathway of molecules to regulate the tolerance of these crops to such adverse growth conditions.

The structural features of the putative CcPMTRs presented in this work, on the one hand, contribute to the near-future study of the role of the key residues highlighted in this work, through site-directed mutations, on the binding properties of melatonin to the receptor and the regulation of its function in plants. On the other hand, they lay the groundwork for future work related to the modulation of the biology of these receptors through the discovery of new selective ligands based on structure-based analysis (molecular docking), where structural adjustment and chemical novelty are prioritized. These receptors could become pharmacological targets to control development and stress tolerance in plants. This has been a successful strategy to regulate important receptors in animals. For example, the docking of over 150 million virtual molecules to the crystal structure of the MT1 melatonin receptor led to the discovery of two selective inverse agonists, which surprisingly advanced the circadian clock phase when administered to mice at dusk [16]. However, a crystal structure of a PMTR is still lacking to further strengthen these studies.

### 3.7. Similarities and Divergence in MTRs from Plants and Animals

Khan et al. [5] suggested a convergent evolution of PMTRs based on a limited similarity in sequence and in the three-dimensional structure with respect to animal GPCRs. However, many elements could be given to argue against. For example, even when sequence similarity is low, key conserved residues participating in the stabilization and/or activation of the receptor still remain.

The activation switch motif in class A GPCRs is NPxxY, while experiments with MTRs involve variations with NAxxY [94]. This motif located in the cytoplasmic end of TM7 participates in key conformational changes associated with GPCR activation [95]. In plants, and after structural alignment with hMT1, the homolog region is LPLLY (L262^7.49At^-Y266^7.53At^), but as a motif it could be Y[LP]PhhY (h stands for hydrophobic), which is well conserved (Figure 10).

Another motif related to Rhodopsin-like receptor family activation is [ED]RY located in TM3 (in cytoplasmic half) involved in structural stabilization of charge and polar interactions especially relevant in maintaining the inactive receptor conformation and G-protein coupling [47,95]. In contrast, MTRs contain the sequence NRY [67]. Although, in plants, a homolog sequence seems to be missing in the transition TM3-ICL2. Of note is that, in TM3, the motif LE[VI]S is almost fully conserved across the proteins analyzed (Figure 10). These changes may indicate a divergence in the activation mechanism of PMTRs.

A YPYP motif present in TM2 is a marker of human MTRs [15], which was linked to receptor stability and shaping ligand binding pockets. The CWxP motif, also known as Na^+^ binding site, contains a W251^6.48^ which is key for bending and rotation of TM6 during receptor activation [47,60]. Additionally, an FIP motif in the vicinity of the former also participates in receptor activation [60,94]. However, none of these were detected in plants, which suggests rearrangements in the architecture of the binding site and the activation mechanism. Alternatively, in the TM2 of PMTRs, the motif QxWEC with great sequence conservation contributes to receptor stability as well. In particular, C35^1.30At^-C99^2.61At^ form a disulfide bridge that is also characteristic of class A GPCRs (C100/C113^3.25^ to C177/C190ECL2, MT1/MT2) (Figure 10).

Clement et al. [96] dilucidated the major role ECL2 plays in MTRs activation. They were able to restore melatonin-induced signaling in GPR50, a closely related orphan GPCR that does not bind MT, after transferring the ECL2 from MT1 to form a chimeric GPR50. This suggests that mutation in this loop may have tremendous effects in PMTRs where more tight or loose ELC2 could impair receptor functionality. A more relaxed or open ECL2 could explain more accessibility to the binding pocket in Clade III receptors, which showed higher melatonin binding capabilities. However, despite such relevance in MTRs function, the amino acidic sequence is poorly conserved in both animals and plants. In PMTRs, F171^5.38At^ is the only residue fully conserved within this loop among spermatophytes (Appendix A).

Additionally, in mammals GPR50 interacts with MT1 and MT2 forming heteromers inhibiting melatonin signaling specifically in MT1 [97]. GPR50 presents only a 45% identity with other hMTRs and has a long C-terminal tail [98]. Interestingly, CcPMTR2 CTD contains a unique extra 46 amino acid long tail which could not be associated to any function after a blastp search.

In spite of the low similarity between animal and plant MTRs, some residues and interactions at the binding pocket appear to be well preserved during evolution. Ligand–receptor interaction in the binding pocket relies mainly on hydrophobic contacts and hydrogen bonds. A structural aligment between MTRs and PMTRs allows us to infer a few similarities, but many residues that are 100% conserved and unique to PMTRs require further exploration into their function. Additionally, PMTRs display unique characteristics, which include an extremely conserved C-terminal region [22].

Note: While this research was being conducted, Chen et al. [99] in a review presented a molecular docking analysis of AtPMTR1 with melatonin in advance. They reported the formation of a subpocket consisting of H36^1.31^, H40^1.35^, E98^2.60^, F171^5.38^, Y250^7.37^, and H258^7.45^, thus indicating the possible existence of another melatonin-binding site on the receptor. The site reported by Chen et al. [99] partially overlaps with site 1 mentioned here. As it was previously mentioned in Section 2.6 residues H36^1.31^, E98^2.60^, F171^5.38^, Y250^7.37^, and H258^7.45^ appear to be important in the interaction with the ligand (Figure 6). Furthermore, in the molecular docking analysis, we demonstrated that F171^5.38^ and Y250^7.37^ interact with the ligand through hydrogen bonds. These coincidences reinforce the existence of another site, different from the one previously reported by Barman et al. [22]. It is important to highlight that the methodological pipeline followed in our study, using a probabilistic approach, allows us to predict the presence of this site in PMTRs with greater robustness and precision. Also, our results from tunnel and cavity analysis indicate that the ligand could access this internal site in an energetically favorable manner, which strengthens the hypothesis that site 1 may be a ligand-binding pocket.

## 4. Materials and Methods

### 4.1. Identification of Melatonin Receptor Candidates

*A. thaliana* PMTR1 (AtPMTR1, Uniprot: Q94AH1 (https://www.uniprot.org/uniprotkb/Q94AH1/entry, accessed on 30 June 2024)) was used as a query to obtain homologous sequences present in the Reference Proteome database using the HMMER web server (http://www.hmmer.org/, accessed on 30 June 2024) [36]. We selected proteins with *E-value* lower or equal than 1 × 10^−100^ and length ranged between 200–400 amino acids belonging to representative species across Viridiplantae [26]. Only the protein with the lowest *E-value* per each species was retained. Inclusion criteria were designed to identify highly homologous sequences while also minimizing information loss and over-representation across taxa. Then, sequence number was enriched by including previously reported PMTR1s from *A. thaliana* [6], *N. benthamiana* [19], *M. sativa* [9], *M. esculenta* [20], *Z. mays* [21], *O. sativa* [22], and *G. hirsutum* [23]. Finally, aligned sequences were trimmed using Gblocks v1.0 [100] and then used to build an HMM using the hmmbuild tool from HMMER package v3.3.2 [101]. Afterward, proteomes from ninety plant species (sensu lato), including major plant orders (Appendix A) were inspected for putative PMTR1 homologs with the HMM previously built. Later, hmmsearch output was parsed using rhmmer package v0.2.0 (https://github.com/arendsee/rhmmer, accessed on 5 July 2024). Ultimately, the identified sequences were further filtered with a more relaxed condition (*E-value* less than or equal to 1 × 10^−20^), and duplicates and partial proteins were removed. Noteworthy, only proteins with seven transmembrane domains were included after a consensus from at least two programs (see below).

### 4.2. Data Download

Metadata including taxa and sequences for each protein entry were downloaded from NCBI (https://www.ncbi.nlm.nih.gov/, accessed on 10 July 2024) or Uniprot (https://www.uniprot.org/, accessed on 10 July 2024) databases using the rentrez package [102] and custom scripts in R v4.4.1. Proteomes from 90 plant species were recovered from databases listed in Appendix A.

### 4.3. Phylogenetic Analysis

Multiple sequence alignment was carried out with Multiple Alignment using Fast Fourier Transform (MAFFT) v7.490 using the L-INS-i method [103]. Later, poorly aligned regions were systematically removed using Gblocks v1.0 with modified parameters for a relaxed trimming with a maximum number of contiguous non-conserved positions (−b3 = 10), minimum length of a block (−b4 = 3) and allowed gap positions (−b5 = h) [104]. Visualization and editing of sequences were carried out with Unipro Ugene v50.0 software [105]. Maximum-likelihood tree was inferred from the resulting alignment using IQ-TREE v2.0.7 [106] where the best substitution model was selected with ModelFinder (−m MFP) [107] and branch support was obtained with UFBoot2 (−B 1000) method [108]. Tree rooting was implemented in treeio package [109] using *Micromonas commoda* homolog sequence as outgroup. Tree annotation and visualization were performed with treeio and ape [110], and ggtree [111] and ggtreeExtra [112] packages, respectively. Species tree was built using OrthoFinder v2.5.5 [27] providing as input 90 plant proteomes (Appendix A) and default parameters. Briefly, multi-copy gene families containing all the species were aligned using MAFFT L-INS-i, then gene trees were built using IQ-TREE [106], later the STAG method [113] used those to build species trees using the “minimum evolution principle”, and finally a consensus species tree was obtained following a “greedy consensus” method. Lastly, these results were cross-validated against Angiosperm Phylogeny Group (APG) (https://www.mobot.org/mobot/research/APweb/, accessed on 12 July 2024) classification.

### 4.4. Topology Prediction and Conserved Motifs

Receptor topology was examined and cross-validated using multiple approaches, including HMM-based methods with Phobius v1.01 [114], as well as protein language models, such as DeepTMHMM v1.0.24 [115] and TMbed [28]. Snake plots were generated using PROTTER v1.0 (https://wlab.ethz.ch/protter/start/, accessed on 1 September 2024) [31]. meme [116] and tomtom [117] tools from MEME suite v5.5.5 [118] were used for motif discovery and for a comparison to ELM (http://elm.eu.org/, accessed on 1 September 2024) and PROSITE (https://prosite.expasy.org/, accessed on 1 September 2024) motif databases, respectively, and their outputs were analyzed with memes package [119] and visualized using universalmotif package [32]. We modified the following meme parameters: distribution of motifs of zero or one per sequence (−mod zoops), minimum (−minw 5), and maximun (−maxw 20) motif width and maximum number of motifs to find (−nmotifs 10). Additionally, regular expressions were designed from relevant motifs to match against protein sequence databases available at Motif server (https://www.genome.jp/tools/motif/MOTIF2.html, accessed on 29 March 2025) and subsequently AmiGO 2 (https://amigo.geneontology.org/amigo, accessed on 29 March 2025) annotations were retrieved using protti v0.9.1 package (https://jpquast.github.io/protti/, accessed on 29 March 2025) [120]. Furthermore, we conducted a protein family classification against the Pfam-A v37.0 (http://pfam.xfam.org/, accessed on 29 May 2024) database [121]. ConSurf server (https://consurf.tau.ac.il/, accessed on 10 March 2025) was used to investigate structurally and functionally evolutionary relevant amino acids using AtPMTR1 sequence as query [29,122,123]. With the purpose of systematizing the naming of residue positions in pPMTRs, in such a way that helps to locate specific residues unequivocally across species, we xtrapolated to PMTRs the “numbering scheme” developed by Ballesteros and Weinstein [30] for GPCRs in animals. Briefly, we selected the most conserved residues after a consensus between ConSurf findings and our results from the alignment of the identified sequences. AtPMTR1 was used as the reference sequence, and the most conserved residue in each TM was selected. The naming is as follows: the first letter denotes the amino acid followed by its position in the protein, and then, in superscripts, the identifier starts with the TM number and ends with its position relative to the reference residue in that TM. That reference residue is arbitrarily assigned the number 50 [30]. To apply to other species, the corresponding residues must be located relative to AtPMTR1 reference residues after an alignment.

### 4.5. Subcellular Localization and Signal Peptide Prediction

Several bioinformatic methods are available for protein localization prediction [124]. We used plant specific homology-based methods with Plant-mSubP (https://bioinfo.usu.edu/Plant-mSubP/, accessed on 10 March 2025) [125] and LOCALIZER 1.0.4 (https://localizer.csiro.au/, accessed on 10 March 2025) [126], which use K-mer compositions and HMM models, respectively. Additionally, a general *ab-initio* method DeepLoc 2.1 (https://services.healthtech.dtu.dk/services/DeepLoc-2.1/, accessed on 10 March 2025) [127] that uses protein language models was also included. We also tested a specific tool developed for *A. thaliana* proteins, AtSubp2 (https://kaabil.net/AtSubP2/prediction.html, accessed on 10 March 2025) [128]. Detection and localization of signal peptides and their cleavage sites was conducted using SignalP 6.0 (https//services.healthtech.dtu.dk/services/SignalP-6.0/, accessed 11 February 2024) [129].

### 4.6. Post-Translational Modifications

Phosphorylation sites, and O- and N-glycosylation were identified with MusiteDeep web server (https://www.musite.net/, accessed on 3 September 2024) [130]. Additionally, phosphorylation sites where also predicted using NetPhos v3.1 (https://services.healthtech.dtu.dk/services/NetPhos-3.1/, accessed on 3 September 2024) [131].

### 4.7. Molecular Docking

Protein structures predicted by Alphafold2 [132] were downloaded from Uniprot (https://www.uniprot.org/, accessed on 10 October 2024). In the case of proteins for which the structure was not yet available, we used Colabfold (https://github.com/sokrypton/ColabFold, accessed on 10 October 2024), which implements a homology search of MMseqs2 with AlphaFold2 to produce faster predictions with similar accuracy [33]. Melatonin and ramelteon 3D structures were downloaded from Pubchem (https://pubchem.ncbi.nlm.nih.gov/, accessed on 10 October 2024). We explored potential docking sites between melatonin and a subset of putative receptors from main taxa groups. Firstly, receptor structures were aligned using UCSF ChimeraX v1.8 [133]. Afterward, a blind semi-flexible docking analysis was carried out using the algorithm implemented in AMDock tool v1.0 [34]. Briefly, ligand and receptor files were prepared via Autodock Tools scripts [134] after adding protonation at pH 7.4. Then, followed a search to identify ligand–receptor binding pockets using Autoligand [135] by exploring the receptor surface to find pockets where interaction energy per volume is maximized. Finally, docking simulations were conducted for each previously detected pocket using Autodock Vina [136]. Ligand efficiency (LE) was estimated using the following equation:$$LE=-\frac{\Delta G}{HA}$$
where ΔG is the free energy of binding and HA is the number of non-hydrogen atoms of the ligand [34]. Additionally, for cross-validation purposes, we used ReverseDock Server (https://reversedock.biologie.uni-freiburg.de/, accessed on 26 February 2025) [137] which implements an inverse blind docking by which one ligand can be tested against various protein targets. Additionally, we selected the two best poses for each protein, defined as those with the lowest Vina scores corresponding to distinct binding pockets. Afterward, to further analyze the score distributions, we performed 1000 simulations using AutoDock Vina, setting the coordinates of each selected pose as the center of a 14 Å grid box. The center of mass for each pose was then calculated by averaging the ligand coordinates. Finally, the resulting centers of mass were classified using the Density-Based Spatial Clustering of Applications with Noise (DBSCAN) algorithm, as implemented in the dbscan package [138] with epsilon (eps = 7) and (minPts = 20) after parameter optimization with Silhoutte scores using cluster [139]. Visualizations were performed in UCSF ChimeraX v1.8 [133].

### 4.8. Docking Site Refinement

AutoDock Vina simulations explore possible ligand poses heuristically, meaning the search for the optimal pose is not exhaustive. To address this limitation and improve the precision of our estimates, we conducted 1000 simulations per site, retrieving only the pose with the lowest binding affinity from each run. The best ligand pose—defined as the one with the lowest binding affinity score—was selected to determine the coordinates for the center of the search space for each site. After performing 1000 simulations, the pose with the lowest score for each site was chosen for further analysis. Prediction of noncovalent interactions was carried out using PLIP server v2.3.0 (https://plip-tool.biotec.tu-dresden.de/plip-web/plip/index, accessed on 1 November 2024) [140]. Additionally, we investigated the effect of missense point mutations in ligand–receptor binding affinity with the alanine scanning approach. For this, we used a machine learning model, PremPLI (https://lilab.jysw.suda.edu.cn/research/PremPLI/, accessed on 1 November 2024) [141].

### 4.9. Tunnel and Ligand Transport Analysis

Receptor surface was explored using CaverDock web server (https://loschmidt.chemi.muni.cz/caverdock/index.html, accessed on 22 November 2024) [142] to detect cavities and tunnels and to track ligand movement toward the binding pocket. Also, Caver Analyst v2.0 BETA software [143] was used to study relevant residues to ligand movement in the entrance and tunnel vicinity. Human MT1 (PDB: 6me2 (https://www.rcsb.org/structure/6me2, accessed on 22 November 2024)) crystal structure was used as a reference where ramelteon position was selected as the starting point to find tunnels toward the receptor surface. Tunnels were considered biologically relevant if: (1) length and bottleneck radius dimensions could accommodate the melatonin and (2) the highest binding energy in the trajectory (Emax, kcal·mol^−1^) was negative. Then, a transport analysis was carried out from the receptor surface to the binding site using default parameters. Afterward, an analogous approach was followed for *A. thaliana* PMTR1 (AtPMTR1), *C. chinense* PMTR1 (CcPMTR1) and PMTR2 (CcPMTR2) receptors bound to melatonin in the lowest binding affinity pose, as determined by docking simulations.

### 4.10. Protein–Protein Interaction Networks

Protein interaction networks were generated for AtPMTR1, CcPMTR1, and CcPMTR2 using STRING server v12.0 (https://string-db.org/, accessed on 13 November 2024) [144]. We recovered both physical and functional associations with interaction scores above 0.4. The interaction network visualization was performed through automation by RCy3 v2.24.0 package [145] using Cytoscape v3.10.3 [146] with the stringApp plugin [147].

### 4.11. Statistical Analysis

Data processing, analysis, and visualization were carried out in RStudio v4.4.1. To test for differences in binding affinity scores between sites for each species, a Student’s *t* test was conducted at a significance level of 95%.

## 5. Conclusions

In this study, we shed light on the evolutionary history of putative plant melatonin receptors through a rigorous identification process and structural and functional characterization derived from models and simulations, the findings of which were supported by cross-validation. We propose a classification of pPMTRs into three major clades and report conserved motifs and residues in these proteins, which could influence melatonin folding and binding stability. We also suggest that these motifs could be involved in receptor activation or signal transduction via protein–protein interactions, being informationally relevant to the development of mutants that characterize receptor functionality and binding capacity. We perform a comprehensive comparison between putative PMTRs and their well-characterized animal counterparts, highlighting key conserved and divergent features. For the first time, an internal binding site for melatonin in plant pPMTRs is proposed, and we provide evidence suggesting that this site is likely the orthosteric binding site, but the existence of secondary sites cannot be ruled out. We expanded the knowledge of this family of receptors using *C. chinense*, a solanaceae highy demanded in the global market. Our results support the relevance of including non-model species with contrasting characteristics, which may reveal functional novelties of these proteins in plants. 

## Figures and Tables

**Figure 10 plants-14-01952-f010:**
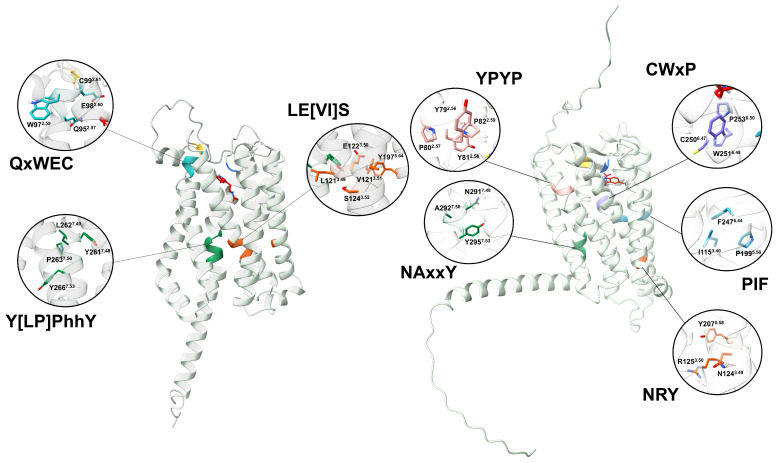
A comparison between melatonin receptors in plants and animals. AtPMTR1 (**left**) and hMT1 (**right**) were used as representatives. Numbering sheme for residues is based on Ballesteros and Weinstein [30].

## Data Availability

The original contributions presented in this study are included in the article/Appendix A. Further inquiries can be directed to the corresponding author.

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
