# Peer review of "Phylogenetic and Structural Insights into Melatonin Receptors in Plants: Case Study in *Capsicum chinense* Jacq"

_plants, 2025, doi:10.3390/plants14131952_

Round 1

Reviewer 1 Report

Comments and Suggestions for Authors

The manuscript entitled “Phylogenetic and structural insights into melatonin receptors in plants: a case study in Capsicum chinense Jacq by Toledo-Castiñeira et al. integrates a complete bioinformatics analysis, in which several melatonin receptor homologs were identified in Viridiplantae proteomes using Hidden Markov Models. Evolutive relationships and conserved domains are indicated. Additionally, docking analysis enabled the identification of melatonin binding sites, where hydrophobic interactions appear to be critical for binding, as well as the establishment of hydrogen bonds. Interestingly, tunnel and ligand transport simulations predicted that melatonin moves through an internal cavity to access the binding site. The authors also predicted protein-protein interactions and discussed their relevance. 

Minor concerns are as follows:

1.- Describe all abbreviations at first mention, reading the document, some of them are fully described in the methods section or on the abbreviation list, but not in the text. If the reader is unfamiliar with specific terms, it becomes difficult to follow.

2.- Figure legend 2: Please describe what is presented in the counts to the right of the tree. This information is only mentioned in the text, but not in the figure legend.

3.- It is suggested that Figure S2A be presented as a principal figure to make it easier to follow the information in Figure 3, especially with respect to the protein topologies displayed next to the phylogenetic tree, regarding the extracellular and intracellular loops. It is also recommended to enhance the quality of Figure 3; when zooming in, the image becomes pixelated, and some data are difficult to read.

4.- Figure S4, indicate site 1 and site 2 in the figure instead of sitio 1 and sitio 2.

5.- The letters in figure S6 are too small and become pixelated when zoomed in.

6.- Figure 5: It would be valuable to add the structure and melatonin binding sites of the human MT1 receptor for comparison. In addition, in the figure 5 legend, is -7 < ΔG  -6 kcal.mol-1correct?

7.- Some scientific names are not italicized (i.e. in figure legend 8); please check the entire text.

8.- Discussion section, lines 358-359, clades II and II. Is it correct?

9.- Section 3.7. Similarities and divergence in MTRs from plants and animals are recommended to be highlighted with a figure or schema. For example, emphasizing the residues and interactions at the binding pocket that are evolutionarily preserved.

Reviewer 2 Report

Comments and Suggestions for Authors

In this study, Toledo-Castiñeira et al. demonstrated that plant melatonin receptors (PMTRs), homologous to animal G protein-coupled melatonin receptors, are widespread across the plant kingdom with significant expansion in angiosperms and can be grouped into three distinct phylogenetic clades. The study identifies 174 putative PMTR homologs across 87 plant species, including Capsicum chinense, and reveals conserved structural features such as seven transmembrane domains and specific conserved motifs, notably an internal melatonin-binding site analogous to that in human MT1 receptors. Molecular docking and binding affinity simulations suggest this internal site has stronger melatonin binding than previously proposed superficial pockets. The structural and phylogenetic analyses provide new insights into the evolution, diversity, and potential functional mechanisms of PMTRs in plants, laying a foundation for future experimental studies to elucidate their role in melatonin signaling and plant physiological responses, including stress tolerance and growth regulation. The study is overall interested however I have some suggestions.

Some suggestions:

  1. How does the expansion of PMTR sequences in angiosperms into three clades (Clade I, II, III) correlate with functional diversification in melatonin signaling across different plant species?
  2. The study predicts differential subcellular localization of PMTRs (e.g., mitochondria in monocots, plastids in eudicots). How might these localizations influence melatonin signaling pathways in response to environmental stresses?
  3. My suggestion is to include mutagenesis studies targeting key residues to validate their roles in melatonin binding and receptor activation. If possible the mutants are ready
  4. The STRING network suggests interactions with G-proteins, kinases, and ubiquitination-related proteins. What are the mechanistic roles of these interactions in melatonin signaling, and how do they differ between clades?
  5. Why do PMTRs in Asterids II show a preference for Clade III sequences, and does this correlate with specific physiological traits or environmental adaptations?
  6. Add in the discussion section by discussing potential biotechnological applications, such as engineering PMTRs for improved stress tolerance in crops, based on the structural insights provided.
  7. Refer to L32, Melatonin receptors (MTRs) has been known in animals for at least 30 years" - should be "have been known" because "receptors" is plural
  8. Refer L51, Plant melatonin receptors (PMTRs) were first reported by Wei et al.6in Arabidopsis (AtPMTR1) who propose a module..." - "who propose" should be "who proposed" to maintain past tense consistency
  9. Refer L51, PMTRs homologs to AtPMTR1 has been characterized in Nicotiana benthamiana..." - should be "have been characterized
  10. For bioinformatics analysis and draw good quality figures please follow these articles doi.org/10.3390/ijms19082216 and doi.org/10.3390/ijms21186624
  11. Refer L90, This suggest that our selection strategy was adequate..." - should be "This suggests..
  12. Refer L42, ECL2 was consistenly the longest domain..." - "consistenly" is a typo; correct spelling is "consistently
  13. Refer L145, again, "consistenly" should be "consistently
  14. "inshights" instead of "insights."
  15. "appropiateness" instead of "appropriateness."
  16. "consistenly" instead of "consistently."
  17. "stablished" instead of "established."
  18. Refer to L142, "ECL2 was consistenly the longest domain with a median of 16 residues." - spelling error and awkward phrasing.
  19. Please upload good-quality figures; in my draft the figures were not very clear.

Reviewer 3 Report

Comments and Suggestions for Authors

Major concerns

  1. Why was Capsicum chinense selected as case? Is it special? In my opinion, the case should be experimental validated. However, the whole contents were in silico analysis. Why other species were not analyzed? It should be meaningful to compare the differences among the three major clades.
  2. How was the phylogenetic tree generated in Figure 2? It should be described in materials and methods or cited related references.
  3. What is the significance of part 2.5 (Post-translational modifications)? Will it affect the binding function of PMTRs?
  4. Figure 5 should be rephrased in an antistic style.
  5. What is the significance of part 2.8? Has the interaction affected the binding function of PMTRs?
  6. The sentences in discussion part should be more cautious since there are no experimental proofs.

Round 2

Reviewer 3 Report

Comments and Suggestions for Authors

The authors have addressed most of the comments. The manuscript could be accepted after grammar checking.